# Retail System Scenario Modeling Using Fuzzy Cognitive Maps

Alina Petukhova *[ID] and Nuno Fachada [ID]

COPELABS, Lusófona University, Campo Grande 376, 1749-024 Lisbon, Portugal; nuno.fachada@ulusofona.pt
* Correspondence: alina.petukhova@ulusofona.pt

**Abstract:** A retail business is a network of similar-format grocery stores with a sole proprietor and a well-established logistical infrastructure. The retail business is a stable market, with low growth, limited customer revenues, and intense competition. On the system level, the retail industry is a dynamic system that is challenging to represent due to uncertainty, nonlinearity, and imprecision. Due to the heterogeneous character of retail systems, direct scenario modeling is arduous. In this article, we propose a framework for retail system scenario planning that allows managers to analyze the effect of different quantitative and qualitative factors using fuzzy cognitive maps. Previously published fuzzy retail models were extended by adding external factors and combining expert knowledge with domain research results. We determined the most suitable composition of fuzzy operators for the retail system, highlighted the system's most influential concepts, and how the system responds to changes in external factors. The proposed framework aims to support senior management in conducting flexible long-term planning of a company's strategic development, and reach its desired business goals.

**Keywords:** retail; complex systems; fuzzy cognitive maps; scenario planning

**MSC:** 97M10; 15B15; 94D05

## 1. Introduction

A retail business can be thought of as a complex system with interconnected components. These include supplier and customer connections, financial and strategic planning, technological operations, staff management, and so on [1]. All these factors have an important role in managerial planning. Intense rivalry, an ever-expanding range of items, numerous consumer segments differentiated by income group, as well as geographic location, and the ability to compete in the mass market, are all characteristics of the retail industry. Managing hundreds of locations around a country, within a constantly changing environment, while ensuring that each one provides the same level of service, is a difficult task. Some of the components of a retail system are tangible, such as profit or advertising costs, while others are intangible, such as employee loyalty or company reputation. Additionally, working with an absolute value of change is impossible, so it is through the weight of a change in a component that its impact on the overall system can be calculated. Component interactions are also nonlinear in time. Customer loyalty, for example, might fluctuate depending on advertising expenses, product costs, and service level; these changes are nonlinear and will affect the system with different time delays. Nonlinear interactions, which represent the influence of the initial conditions from which this or that form of action in a developed complex system, make statistical analysis of these processes extremely difficult [2]. To simulate prospective behavior change, managers undertaking scenario planning and strategy analysis for an organization must examine the system's history and current status.

Strategic planners frequently struggle to grasp the complex chains of cause and effect in their changing environment and their impact on planned strategy. They are confronted with challenges that cannot be solved by simply collecting more data, clarifying problems,

or dividing them into smaller ones [3]. For a long time, modeling has been one of the most popular methods for management planning [4]. It helps decision-makers to analyze complex problems, allowing them to plan future strategies for the organization more scientifically and objectively.

Developing retail models requires the type of specialized knowledge which is easily described with fuzzy cognitive maps (FCMs). Since FCMs have been applied for strategic planning process simulation [5], and to understand the impacts of different actions on revenue and costs [6], we propose the application of this approach to develop a more specialized model of retail business. FCMs can handle complexity as well as incomplete or conflicting information for ill-structured systems. FCMs were first proposed at 1986 by Kosko [7] and represent a system as a causality graph that comprises *concepts* and *links* between them.

This paper focuses on the three main research questions: (1) Is it possible to extend a retail FCM by including external economical and political factors? (2) How do we choose mathematical methods to improve the quality of FCM models? (3) What kind of forecasts can we make about a retail system modeled with FCMs?

To create a fuzzy model of a retail system and include external factors, we reviewed various components of a retail organization, including interaction with consumers and staff, relationships with suppliers, market conditions, and political stability of the country. The developed FCM model allows reduction of uncertainty in the decision-making process, significantly decrease in scenario planning time, and allows anticipation of the effect of changing multiple factors on the financial performance of the company. The model also supports the analysis of extreme external factor changes, such as political or economic crises, promoting the development of appropriate responses.

Integrating FCMs in retail business planning enables the analysis of many alternative scenarios in a short time frame. This would reinforce planning frequency, improve the quality of forecasts, decrease managerial risk, and rapidly respond to environment changes by adding external factors to the FCM. With a mathematical model of business process, stakeholders can change one or more factors and assess the future potential impact on the entire retail business.

This paper is organized as follows. The backdrop of retail strategic planning, existing models, and data sources used in the sector are presented in Section 2. Section 3 informs on how we developed the retail system's FCM model. In Section 4, we describe the main findings and results of the study, which are in turn interpreted and discussed in Section 5. Finally, in Section 6 we provide the conclusions and suggested future studies.

## 2. Background

Companies have been seeking to improve managerial processes as a result of today's competitive business climate. The globalization of national economies has led to the rapid expansion of retail markets, both in terms of reaching more communities and variety of products and services available. Models help managers to build sustainable corporate development strategies and examine the impact of decisions on all company operations.

In 1998, Lee et al. [8] presented a mechanism for integrating fuzzy cognitive map knowledge with a strategic planning simulation. This mechanism helped to understand how complex chains of influence under a changing environment impact the performance of implemented strategies. Later, several studies suggested using different variations of FCM during company strategical planning [9–11]. In reference [10], Lee proposed an agent-based inference method to overcome FCM dynamic relationships, time lags, and re-usability issues of FCM evaluation in strategic planning. The utilization of FCM in the modeling of business and management systems has been presented in [6].

The retail market interpreted in terms of complex systems was discussed by Pennacchioli et al. [12]. This approach helped uncover emerging regularities and patterns that make markets more predictable by estimating how much a country's GDP will grow. The authors examined a unique transaction database containing the micro-purchases of a

million customers in the stores of a national supermarket chain for several years, revealing fundamental patterns which connect the product volumes of sales with the volume of purchases made by customers.

The retail sector has many specific characteristics that have to be considered during the modeling process, such as intense pressure of competition, a steadily changing portfolio of products, and ever-changing customer requirements. A number of researchers have examined components of the retail system in their studies. For example, Sadler [13] focuses on optimizing decisions on the siting of a planned healthy food retail intervention; Nagibina [14] has analyzed the labor efficiency in food retails; Zhosan and Kyrychenko [15] have discussed a comprehensive system for evaluating the efficiency of the retailer's pricing policy.

In another group of publications, the authors have analyzed employee loyalty in retail [16–18]; later, Silvestro and Cross [19] examined customer service levels in companies, and Veloso and Monte [20] discussed the effect of perceived value and quality service on customer satisfaction. Recently, Winkler et al. [21] reviewed the use of complex system models in retail food systems with the purpose of boosting population health. The latest trends of moving to online channels in retail were described by Reinartz et al. [22], Ariannezhad et al. [23], and Wu [24].

In this paper, we focus on retail businesses with physical stores, and analyze retail business components by representing them in an FCM structure. While FCM models were previously used for offline retail systems [25,26], this work aims to extend existing literature by adding more complex internal and micro-economical concepts, while providing a more detailed mathematical formulation of the model. Moreover, we describe the effect of fuzzy logic operator selection on the overall system, apply system indicators to enhance the analysis of retail FCMs, and suggest potential modeling scenarios for future studies.

## 3. Methods

This section describes the mathematical formulation of the proposed retail system FCM, the system indicators used for developing the FCM analysis, as well as the approach for constructing the model using expert interviews and data from domain research articles. Section 3.1 provides a brief introduction to the theory of FCMs, as well as the basic equation that describes the system and is employed in scenario modeling. The implemented FCM is assessed using system characteristics, which are explained in Section 3.2, establishing a level of confidence in the model prediction and determining its most essential components. The steps for building a retail scenario FCM, as well as the methods for combining the different data sources used in this work, are described in Section 3.3. Finally, a software implementation of the proposed methods is presented in Section 3.4.

### 3.1. FCM as a Modeling Approach

FCM is a soft computing method for modeling, analyzing, and describing complex causal systems. As first proposed by Kosko [7], an FCM is a knowledge graph made up of (1) nodes representing concepts, and (2) relationships between them represented by links. Link weights are defined in the range $[0, 1]$ if negative influences are not considered in the model, or $[-1, 1]$ otherwise. Expert knowledge, research data, statistical aspects of the system, and/or historical data from the system are used to construct concepts and link weights.

An FCM model can be defined by $\mathbf{A}^{(0)}$, the vector containing the initial state of the nodes, and $\mathbf{W}$, the matrix of link weights. The state $A_i^{(t)}$ of the concept $i$ at time $t$ can be calculated as the combination of the influences of all concepts linked to $A_i^{(t)}$ and is represented by Equation (1). This equation describes the activation rule for FCMs, where the state characterizes the degree of activation of the concept at each time step.

$$A_i^{(t+1)} = f\left( \sum_{\substack{j=1 \\ j \neq i}}^{N} w_{ji} A_j^{(t)} \right) \tag{1}$$

Here, $w_{ji}$ is the weight of the link between concepts $j$ and $i$, $A_j^{(t)}$ is the state of concept $j$ at the time $t$, and $f(\cdot)$ is the activation function, $N$ is the number of the concepts in FCM. The most common activation function is the sigmoid, $f(x) = \frac{1}{1+e^{-\lambda x}}$, which maps its input to the $[0,1]$ interval with steepness $\lambda$ (with $\lambda > 0$, so that the function's output monotonically increases).

Alternatively, the composition of the link weights and concept states can be represented using any disjunctive *S*-norm and conjunctive *T*-norm fuzzy operators [27]. This, Equation (1) can be re-written as

$$A_i^{(t+1)} = S_{\substack{j=1 \\ j \neq i}}^{N}\left(T(w_{ji}, A_j^{(t)})\right) \tag{2}$$

where $T$ is the triangular norm operation, and $S$ is the triangular conorm operation. *T*-norms and *S*-norms are the causal algebra operators used to solve the problems of, (1) accumulating the influence of several governing concepts on a target concept, and, (2) determining the indirect influence of concepts.

After a change in one of the concepts, the FCM will provide an output containing information on the degree of activation of all concepts at each discrete time step. This cycle repeats itself until a certain termination condition is fulfilled. After several iterations, it can converge to a balance point, a chaotic point, or an intermittent attractor. Individual fuzzy influences of input concepts that directly affect the output concept are integrated into an FCM pairwise, starting from the first concept using the *S*-norm, which is specified on the $[0,1]$ interval. The concept state and weight of the influence are linked with the *T*-norm. The most common operator is the *minimum*, which is used as an intersection operation, and the dual *S*-norm operator *maximum*, which represents the sum operation. Several commonly used *T*-norms and *S*-norms [28] are given in Table 1.

**Table 1.** Commonly used *T*-norms and *S*-norms, where $p$ and $q$ are the state of the concepts $j$ and $i$, respectively.

| *T*-Norm | *S*-Norm |
|---|---|
| *Minimum* $T_M(p,q) = \min(p,q)$ | *Maximum* $S_M(p,q) = \max(p,q)$ |
| *Algebraic product* $T_P(p,q) = pq$ | *Algebraic sum* $S_P(p,q) = p + q - pq$ |
| *Hamacher product* $T_H(p,q) = \begin{cases} 0, & \text{if } p = q = 0 \\ \frac{pq}{p+q-pq}, & \text{otherwise} \end{cases}$ | *Hamacher sum* $S_H(p,q) = \begin{cases} 0, & \text{if } p = q = 0 \\ \frac{p+q-2pq}{1-pq}, & \text{otherwise} \end{cases}$ |
| *Einstein product* $T_E(p,q) = \begin{cases} 0, & \text{if } p = q = 0 \\ \frac{pq}{2-(p+q-pq)}, & \text{otherwise} \end{cases}$ | *Einstein sum* $S_E(p,q) = \begin{cases} 0, & \text{if } p \cdot q = -1 \\ \frac{pq}{1+pq}, & \text{otherwise} \end{cases}$ |
| *Drastic product* $T_D(p,q) = \begin{cases} max(p,q), & \text{if } min(p,q) = 0 \\ 1, & \text{otherwise} \end{cases}$ | *Drastic sum* $S_D(p,q) = \begin{cases} min(p,q), & \text{if } max(p,q) = 1 \\ 0, & \text{otherwise} \end{cases}$ |
| *Nilpotent minimum* $T_N(p,q) = \begin{cases} \min(p,q), & \text{if } p + q \geq 1 \\ 0, & \text{if } p + q < 1 \end{cases}$ | *Nilpotent sum* $S_N(p,q) = \begin{cases} \max(p,q), & \text{if } p + q < 1 \\ 0, & \text{if } p + q \geq 1 \end{cases}$ |
| *Lukasiewicz max* $T_L(p,q) = \max(p + q - 1, 0)$ | *Lukasiewicz min* $S_L(p,q) = \min(p + q, 1)$ |

With respect to *T*-norms, the use of the *minimum* operator $T_M$ can have advantages for systems in which the information processing method is close to boolean, i.e., most of the dependencies between the input and output values of the system are of a logical

nature. Thus, the scope of the *minimum* operator $T_M$ in FCM is limited. The advantage of the *algebraic product* operator $T_P$ is that its output value has a quantitative dependence on the actual values of both parameters, except for the case in which one of parameters is zero. Of course, the loss of information for this operator is not as significant as for $T_M$, in which the value depends only on the smallest component. The retail system model developed in this work was tested using different fuzzy operators to identify the most suitable for the given task. Consequently, we propose a rule to select the best $T$-norm and $S$-norm operators for the system that will be described in Section 4.

Equation (2) describes an update rule that is widely used in many FCM-based frameworks for scenario modeling. However, to account for concept activation values in the previous step, Stylios and Groumpos [29] proposed a modified update rule, as shown in Equation (3). With this rule, concepts, in addition to the corresponding weights and activation values of other concepts, take into account their past activation values. This update rule is preferred when the system contains weekly connected concepts that are not affected by many other concepts, and is more relevant for systems with many components, such as retail systems.

$$A_i^{(t+1)} = S_{\substack{j=1 \\ j \neq i}}^{N}\left(T(w_{ji}, A_j^{(t)}), A_i^{(t)}\right) \tag{3}$$

Link weights may take positive and negative values. For example, in retail systems, a price increase will cause a decrease in customer loyalty [30]. To add negative causal links between FCM concepts, we used the algorithm described by Silov [31]. Its main idea is to create a second matrix describing all negative influences in the system. To do so, the initial dimensions of the FCM's weight matrix are doubled by separating positive and negative influences using the rule described by Equation (4) to obtain matrix **R** containing only positive weights. Considering matrix **W** is of size $N \times N$, $N$ being the number of concepts considered in the model, matrix **R** will be of size $2N \times 2N$.

$$\begin{cases} r_{2i-1,2j-1} = w_{ij}, r_{2i,2j} = w_{ij}, \text{if } w_{ij} > 0 \\ r_{2i-1,2j} = -w_{ij}, r_{2i,2j-1} = -w_{ij}, \text{if } w_{ij} < 0 \\ r_{2i-1,2j-1} = 0, r_{2i,2j} = 0, \text{if } w_{ij} = 0 \end{cases} \tag{4}$$

Since the weight of the links in FCMs generally depends on the expert opinion and is not precise, some of the links may be missing, and/or their weight may not be defined accurately. To adjust the weights matrix and identify the hidden connections between concepts we used the probabilistic transitive closure (PTC) approach [32], which consists of computing a bipolar weighted digraph with the same set of concepts as the initial FCM, but with links corresponding to the indirect impacts in the given FCM. The weight of a link $(r_i^*, r_j^*)$ in the PTC matrix **R**$^*$ is the probability of the event that the FCM has a directed walk from $r_i^*$ to $r_j^*$. The PTC matrix is obtained from matrix **R** with Equation (5):

$$\mathbf{R}^* = \mathbf{R} \circ \mathbf{R} \tag{5}$$

where $\circ$ is the selected $S$-norm.

After computing indirect impacts in the FCM with PTC, matrix **R**$^*$ can be reduced to the initial size $N \times N$ in the form of matrix **V**, which is generated using the transformation described in Equation (6):

$$\begin{cases} v_{ij} = \max(r_{2i-1,2j-1}, r_{2i,2j}) \\ \overline{v}_{ij} = -\max(r_{2i-1,2j-1}, r_{2i,2j}) \end{cases} \tag{6}$$

where $v_{ij}$ and $\overline{v}_{ij}$ represent positive and negative influence of the concepts, respectively. The elements of the resulting matrix **V** are used in the scenario modeling and allow to carry on problem-targeted analysis in complex systems.

### 3.2. System Indicators

System indicators are commonly used methods for measuring uncertainty in FCMs. They help to characterize the system and analyze its main properties, such as the impact of different concepts and the level of trust or distrust in model predictions. Several system indicators are presented in Table 2. The most common system indicators in FCMs are concept's consonance and dissonance, the impact of concepts on the system ($\vec{P_i}$), and vice versa ($\overleftarrow{P_j}$) [33].

**Table 2.** System indicators used in FCMs.

| Indicator | Formula |
|---|---|
| Concept's consonance | $c_{ij} = \dfrac{\|v_{ij} + \bar{v}_{ij}\|}{\|v_{ij}\| + \|\bar{v}_{ij}\|}$ |
| Concept's dissonance | $d_{ij} = 1 - c_{ij}$ |
| Concept's impact on the system | $p_{ij} = sign(v_{ij} + \bar{v}_{ij})max(\|v_{ij}, \bar{v}_{ij}\|),$ $v_{ij} \neq -\bar{v}_{ij}$ |
| Consonance of the i-th concept influence on the system | $\vec{C_i} = \frac{1}{n}\sum\limits_{j=1}^{n} c_{ij}$ |
| Consonance of the system's influence on the j-th concept | $\overleftarrow{C_j} = \frac{1}{n}\sum\limits_{i=1}^{n} c_{ij}$ |
| Dissonance of the i-th concept influence on the system | $\vec{D_i} = \frac{1}{n}\sum\limits_{j=1}^{n} d_{ij}$ |
| Dissonance of the system's influence on the j-th concept | $\overleftarrow{D_j} = \frac{1}{n}\sum\limits_{i=1}^{n} d_{ij}$ |
| Impact of the i-th concept on the system | $\vec{P_i} = \frac{1}{n}\sum\limits_{j=1}^{n} p_{ij}$ |
| Impact of the system on the j-th concept | $\overleftarrow{P_j} = \frac{1}{n}\sum\limits_{i=1}^{n} p_{ij}$ |

Matrices $\mathbf{C_s} = [c_{ij}]$, $\mathbf{D_s} = [d_{ij}]$, $\mathbf{P_s} = [p_{ij}]$ are called cognitive matrices of joined consonance, joined dissonance, and joined impact of the system, respectively. In the model, these matrices are used to find the groups of concepts that strengthen or weaken the system ($\mathbf{P_s}$), or define the level of confidence in the plus ($\mathbf{C_s}$) or minus ($\mathbf{D_s}$) sign of the cognitive relationship. As the value of the consonance increases, the confidence in the sign of influence increases. By setting up a relevancy threshold in matrix $\mathbf{P_s}$, the most influential concepts in the system can be identified.

Vector $\overleftarrow{\mathbf{P}}$, given by $\overleftarrow{\mathbf{P}} = \begin{bmatrix} \overleftarrow{P_1} & \overleftarrow{P_2} & \dots & \overleftarrow{P_n} \end{bmatrix}$ (note that this equates to the sum of the columns of $\mathbf{P_s}$), locates concepts that are highly impacted by the system and are harder to control. The analysis of vectors $\vec{\mathbf{P}} = \begin{bmatrix} \vec{P_1} & \vec{P_2} & \dots & \vec{P_n} \end{bmatrix}$ and $\vec{\mathbf{C}} = \begin{bmatrix} \vec{C_1} & \vec{C_2} & \dots & \vec{C_n} \end{bmatrix}$ determines the concepts that are stabilizing the system and can change its dynamic (note these vectors are also given by the sum of the rows of the respective matrices). Higher values in $\vec{\mathbf{P}}$ correspond to the most influential concepts, i.e., a change in these concepts will have the strongest effect on the model. An additional factor to analyze is the trust in the concept value, represented by $\vec{\mathbf{C}}$. Concepts with the highest values have the most confidence in the direction (sign) of influence. In conclusion analysis of the system indicators provides the ability to examine the system's development, stability, and sensitivity to changes.

### 3.3. Model Creation

We begin by selecting the defining concepts of the system based on previous research and expert interviews. The next step is to characterize the relationships between concepts as the weighted average of experts' estimates. Finally, the weights of the relationships are

adjusted by using the previous research data. The exact process of model creation depends on the number of experts and available information about the system and is detailed next for the case of a retail system model.

At the first stage of building the model, the general factors that determine the retail processes were identified. Model concepts were derived from existing models [34], strategic planning guidelines [35], and general descriptions of the retail industry [36,37]. It is vital when modeling any complex system to define system concepts and their interactions appropriately.

Since some of the concepts are categorical variables, the relationship between them cannot be extracted from historical system data. To define these relationships, three experts in the retail sector were interviewed. The experts have different tenure in the area and are in positions of business analyst, senior business analyst, and department head. With the linguistic equivalents "Very low", "Low", "Medium", "High", "Very high", and the direction of influence, positive or negative, experts were asked to assess the influence of one concept on another. Then, the linguistic equivalents were converted into numerical values (0–4) and combined using a weighted average mean based on the tenure of the expert [38]. With the experts interviews, we can identify concept relationships that are difficult to measure and are not influenced by statistical analysis such as "Customer satisfaction" or "Company reputation".

In the next step, a number of research articles were analysed in order to obtain further information about concept relationships. These publications, and the derived concept influences, are as follows:

- Publications on the employee loyalty topic [16–18]:
    - "Employee loyalty"—"Staff turnover"
    - "Employee loyalty"—"Profit"
    - "Employee loyalty"—"Customer service level"
- Publications on the customer service topic [19,37]:
    - "Customer service level"—"Profit"
    - "Customer service level"—"Customer loyalty level"
- Publications on the staff turnover topic [16,20]:
    - "Staff turnover"—"Working conditions"
    - "Staff turnover"—"Cross department support"
    - "Staff turnover"—"Openness of communication with employees"
    - "Staff turnover"—"Profit"
- Publications on the company profit topic [39]:
    - "Profit"—"Bank loan"
    - "Profit"—"Fixed assets"
    - "Profit"—"Working capital"
    - "Profit"—"Stock prices"

Equal weight was given to expert knowledge and publication-derived influences in order to combine them into a final set of concept relationships. Anonymized expert interviews, publication-based influences, and aggregation results are publicly available at the Zenodo open-access scientific repository [40].

Next, concepts were divided into seven groups (or subsystems) based on the strongest influence between them, namely: technology, employees, finance, customers, external factors, suppliers, and investments. These groups, as well as the associated concepts, are presented in Table 3.

Each subsystem can influence the choice of strategy, as well as the final profit and development path of the company, and should be taken into account during scenario planning. Thus, the uncertainty in explaining the behavior of the retail system can be reduced. The final system is shown in Figure 1 in the form of an FCM graph. The complete initial matrix underlying this graph is available in the supplementary materials [40], as well as the data supporting the results presented in the next section.

**Table 3.** Concepts and subsystems defined for the retail FCM. The K* column denotes concept identifiers.

| Subsystems | Concepts | K* |
|---|---|---|
| Technology | Technical level of equipment | K1 |
| | Production standards | K2 |
| | Speed of adoption of innovative technology | K3 |
| | IT infrastructure | K4 |
| Employees | Number of trained staff | K5 |
| | Lost working time | K6 |
| | Labor productivity | K7 |
| | Working conditions | K8 |
| | Employee loyalty | K9 |
| | Staff turnover | K10 |
| | Cross department support | K11 |
| | Openness of communication with employees | K12 |
| Finance | Market competition level | K13 |
| | Interest rate on loans | K14 |
| | Accounts payable | K15 |
| | Bank loan | K16 |
| | Amount of taxes paid | K17 |
| | Sales revenue | K18 |
| | Profit | K19 |
| | Total costs | K20 |
| | Fixed assets | K21 |
| | Rent | K22 |
| | Advertising costs | K23 |
| | Currency exchange rate | K24 |
| | Market share | K25 |
| | Working capital | K26 |
| | Stock prices | K27 |
| Customers | Customer demand | K28 |
| | Customer income level | K29 |
| | Product quality | K30 |
| | Customer service level | K31 |
| | Customer loyalty level | K32 |
| | Company reputation | K33 |
| | Assortment of goods | K34 |
| | Margin on goods | K35 |
| | Price segment of goods | K36 |
| | Share of the internal branded goods | K37 |
| External factors | Political stability | K38 |
| | Inflation expectations | K39 |

**Table 3.** *Cont.*

| Subsystems | Concepts | K* |
|---|---|---|
| Suppliers | Supplier' purchase price | K40 |
| | Supplier' purchase terms | K41 |
| | Effectiveness of supplier selection | K42 |
| | Supplier' technical readiness | K43 |
| | Integration of systems with suppliers | K44 |
| Investments | Domestic investments | K45 |
| | Capital investments | K46 |
| | Foreign investment | K47 |

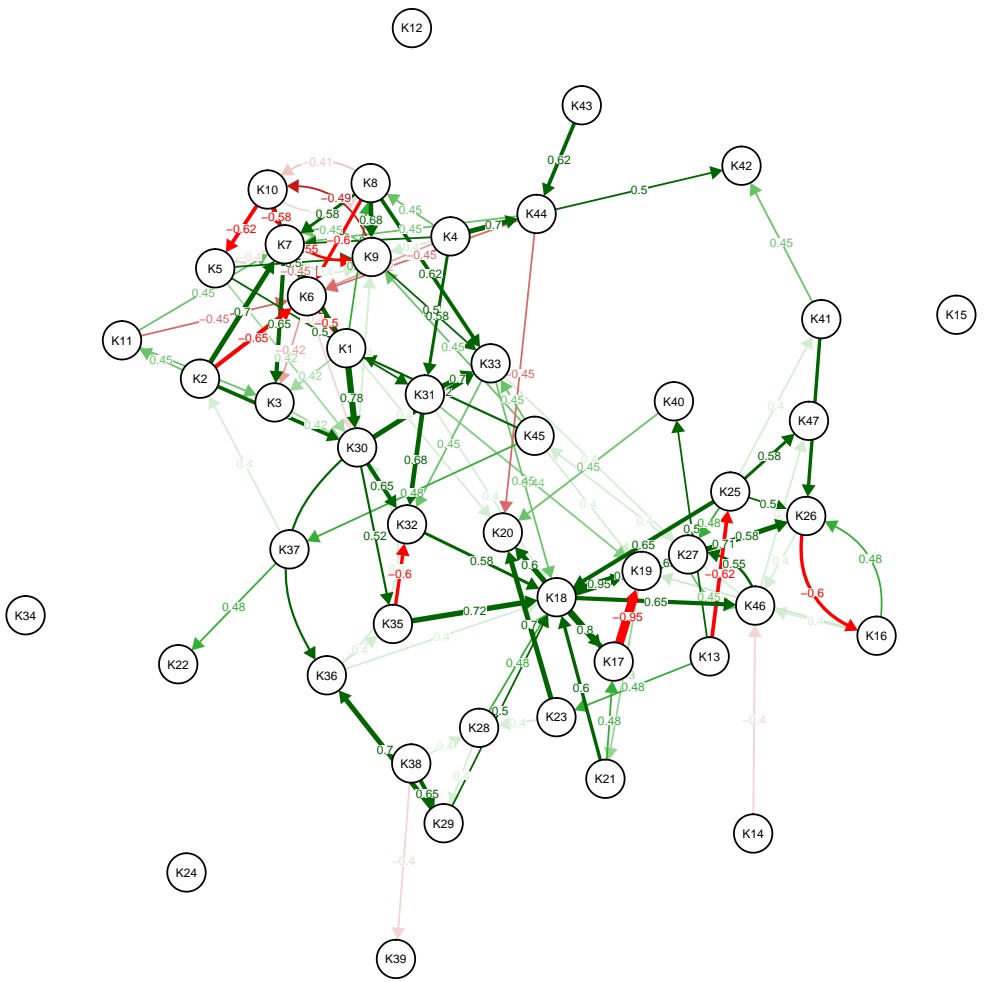

**Figure 1.** Fuzzy cognitive map of the retail system. Connections are only shown for absolute influences equal or greater than 0.4. Green arrows represent positive influences, while red arrows show negative influences. Thicker arrows imply stronger concept influences. Isolated concepts have absolute influences below 0.4.

### 3.4. Software Implementation

The described theory is implemented in the form of an R package at https://CRAN. R-project.org/package=FuzzyM (accessed on 16 February 2022) and is fully open source under the MIT license, which allows liberal use of the code.

## 4. Results

The initial FCM (Figure 1) was trained using the PTC method. The process of choosing a fuzzy operator for training is described in Section 4.1. A structural analysis of the trained model is undertaken in Section 4.2, where we highlight some of the system's main properties. Finally, in Section 4.3 we discuss three modeling scenarios for the retail system, together with a detailed analysis of one of these scenarios.

### 4.1. Fuzzy Logic Operator Selection

Training an initial FCM allows the discovery of hidden links between concepts, and a fuzzy composition operator is required for this purpose. Optimistic forecasts can be generated with the fuzzy *T*-norm *Minimum* and *S*-norm *Drastic sum* operators combination and pessimistic forecasts with the *T*-norm *Drastic product* and *S*-norm Maximum operators. The most widely utilized composition operator combination in the literature is *T*-norm *Minimum* and *S*-norm *Maximum* [2], which is employed when a system requires conservative solutions, i.e., when the quality of one concept cannot compensate for the poor quality of another. However, in real systems some situations allow you to compensate for the values of the input vector. In this case, the liberal *Minimum T*-norm is not the best choice for the intersection of fuzzy sets. Figures 2 and 3 represent consonance of the overall influence between concepts using different *T*-norm and *S*-norm operators. Analysing these figures, it can be seen that different composition operators generate distinct clusters and connections between concepts. The operator combination *T*-norm *Minimum* and *S*-norm *Maximum* [2] is not suitable for the created retail FCM, since it adds no additional value to the analysis due to imposing stringent limits on the system, as shown in Figure 3.

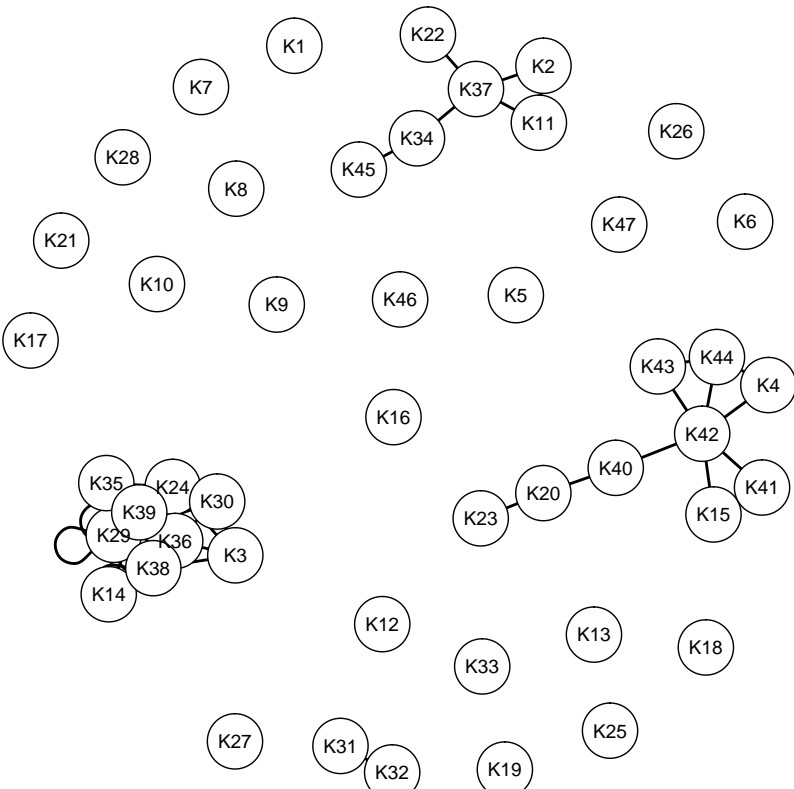

**Figure 2.** Consonance of the overall impact of concepts using the operator combination *T*-norm *Algebraic product* and *S*-norm *Maximum* and relevancy threshold of 0.85.

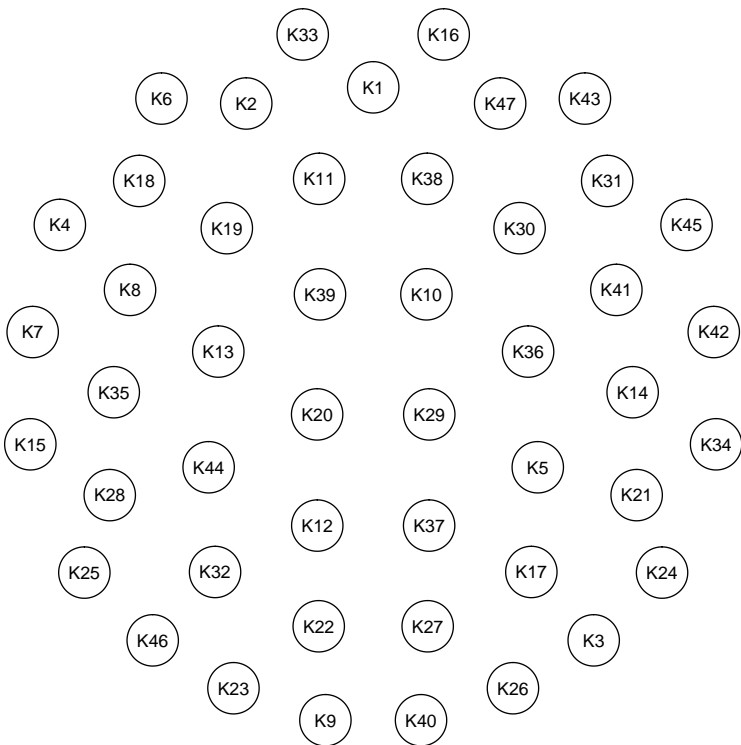

**Figure 3.** Consonance of the overall influence of concepts when using the operator combination *T*-norm *Minimum* and *S*-norm *Maximum* and relevancy threshold of 0.85.

References [41,42] provide some suggestions for selecting an effective fuzzy composition operator. We propose a selection rule based on the elbow method [43], commonly used for determining the number of clusters in a data set. The basic idea behind the method is to graph the number of unrelated concepts for the consonance indicator and calculate the best composition operator based on the angle of the curve, as shown in Figure 4 for the case of the proposed retail FCM. The chosen operator will correlate to the curve's elbow point.

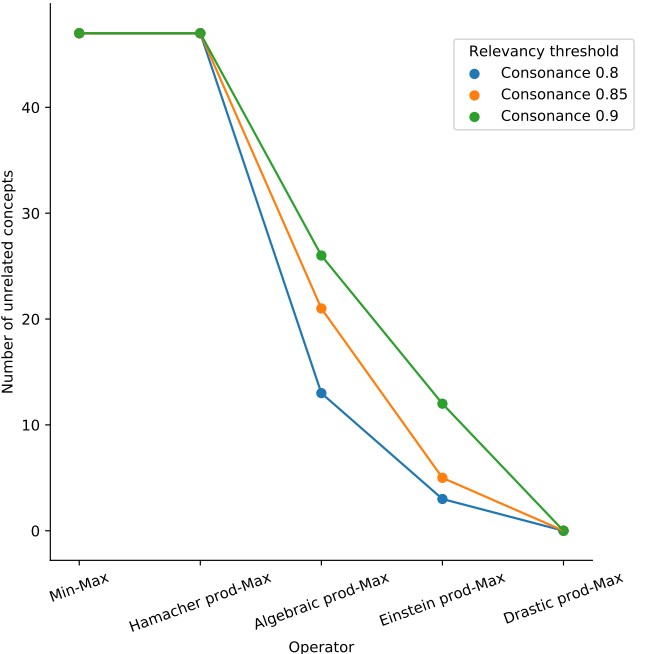

**Figure 4.** Number of unrelated concepts for the retail FCM when using different composition operators and relevancy thresholds.

For the retail system considered in this work, the composition of the operators *T*-norm *Algebraic product* and *S*-norm *Maximum* is preferred since it can provide superior results, as shown in Figure 4. As can be seen, this operator combination is optimally located between the fully unconnected FCM given by the *Minimum–Maximum* and *Hamacher product–Maximum* operators, and maximally connected FCM generated with *Drastic product–Maximum*. With the *Algebraic product–Maximum* operator, it is possible to build clusters and evaluate the performance of the system while not complicating its analysis by introducing a high number of clusters.

### 4.2. Structural Analysis of the FCM

The structural analysis of an FCM promotes the discovery of inconsistencies in the original weight matrix, $\mathbf{W}$, i.e., inconsistencies in the weights of causal connections. Therefore we performed a structural analysis of the created FCM's attributes, which resulted in the identification of several positive feedback loops represented by concepts that have a beneficial influence on each other. For example, in the employees relationships subsystem, "Cross department support" (K11) increases the "Labor productivity" (K7) concept. This can be interpreted as the simplification of the processes that allow employees to focus on current tasks, which later helps to increase "Employee loyalty" (K9) by assisting employees in completing tasks on time, which strengthens K11 as a practice in the company.

A retail business is a self-adaptive system that can modify behavior based on the present configuration and perception of the surroundings. The system responds to changing environmental conditions by developing new states. This can be observed in the interaction between "Interest rate on loans" (K14), "Inflation expectations" (K39), and "Political stability" (K38). Reducing the "Interest rate on loans" (K14) raises "Inflation expectations" (K39), and the stronger the country's inflationary expectations, the less "Political stability" (K38) it would have. A reduction in the country's "Political stability" (K38) will require an increase in the "Interest rate on loans" (K14). In some scenarios, such as increased inflation, a company needs a combination of a financial plan—to address increased expenses and a reduction in demand—and a political response. The latter can consist of raising the amount of self-produced commodities, price reductions, or increasing profits through improvements in customer satisfaction.

We inspected several system indicators, such as the impact of concepts on the system ($\overrightarrow{\mathbf{P}}$), system's impact on concepts ($\overleftarrow{\mathbf{P}}$), consonance ($\overrightarrow{\mathbf{C}}$ and $\overleftarrow{\mathbf{C}}$), and dissonance ($\overrightarrow{\mathbf{D}}$ and $\overleftarrow{\mathbf{D}}$). The analysis of the vectors $\overrightarrow{\mathbf{P}}$ and $\overleftarrow{\mathbf{P}}$ shows that the system reinforces the concept of "Company reputation" (K33), at the same level that the concept reinforces the system. Since the dissonance of this concept is below average, there is a tendency for the company's reputation to stabilize. The system marginally strengthens the concept "Profit" (K19), but its consonance is below average, reflecting the pessimistic tendency of the concept development due to the low level of trust. The model demonstrates that, without any changes to the system, retail profits will tend to decline, which can be explained by severe competition and poor margins in the retail sector.

"Customer service level" (K31), in contrast with other concepts, enhances the system the most, having a positive influence of 0.16, while the system's influence on the concept is, in turn, weaker. This underscores the notion that "Customer service level" (K31) will always be an important indicator of how people view retail brands at the system level. Customers return to stores after having a pleasant shopping experience; thus, processes must be in place to provide a consistent and customer-focused experience. Since the system's and concept's consonance are nearly the same, we conclude that K31 strengthens the system.

The concepts "Product quality" (K30) and "Production standards" (K2) are next in terms of the degree of positive strengthening (impact) of the system. "Sales revenue" (K18) and "Employee loyalty" (K9) grow as a result of the system's impact, whereas these concepts have less impact on the system. In comparison with other concepts, the system weakens the concept "Staff turnover" (K10) the most, since it has the lowest value in $\overleftarrow{\mathbf{P}}$. The "Interest

rate on loans" (K14) concept has a negative impact on the system, while the K14 itself is weakened by the system. The system's consonance and the concept's consonance are in the same range, thus demonstrating how vulnerable the retail sector is to interest rate increases. A rise in interest rates will directly affect mortgage rates for the consumers, which may lead to a reduction in everyday spending, while retail companies may deepen their book debt due to current loans. The "Bank loan" concept (K16) is debatable: the system weakens it, yet it has a minor positive effect on the system.

Clusters of consonance, dissonance, positive and negative impacts were used to continue the FCM analysis. To generate the clusters we utilized various operators of causal algebra composition with a defined relevancy threshold. This method allows us to identify groups of similar concepts and analyze the relationships within the clusters of interest.

Elements that have a substantial influence both within a subsystem and on the entire system can be distinguished in each cluster, as well as the factors that are less important and have weak connections. For example, the consonance plot presented in Figure 2 shows the concepts that form positive or negative feedback loop clusters with a relevancy threshold of 0.85. From this plot we can highlight the following clusters and suggest their possible meaning:

1.  "Customer service level" (K31), "Company reputation" (K33), "Product quality" (K30), and "Technical level of equipment" (K1) form a cluster of customer security in the market.
2.  "Working capital" (K26), "Profit" (K19), "Sales revenue " (K18), "Margin on goods" (K35), "Amount of taxes paid" (K17), "Total costs" (K20), and "Advertising costs" (K23) form a financial cluster.
3.  "IT infrastructure" (K4), "Integration of systems with suppliers " (K44) form an IT cluster.
4.  "Customer income level" (K29), "Price segment of goods" (K36) are combined into a customer cost sensitivity cluster.
5.  "Production standards" (K2), "Labor productivity" (K7) constitute a production cluster.

Figure 5 presents the mutual influence of system elements when using the composition *T*-norm *Algebraic product* and *S*-norm *Maximum*, in which a cluster of concepts with positive impact can be highlighted: "Product quality" (K30), "Technical level of equipment" (K1), "Company reputation" (K33), and "Customer service level" (K31). Since K31 can be directly controlled by the retail company, it can be critically important to improve its quality to increase related concepts of the cluster. Amazon's customer service is a good example of a customer service-focused business approach, with the buy and return procedure tailored to reduce customer effort and maximize response times [44].

*4.3. Scenario Modeling with FCM*

In this section we discuss three proof-of-concept scenarios with high-level analysis of the first two scenarios and detailed analysis of the third one.

In the first scenario, and in order to examine the system's response to external input, we defined a group of concepts whose states were modified, and traced the internal system state change over finite time intervals. In this scenario, we considered the increase of "Integration of systems with suppliers" (K44). This action led to the reduction of "Total costs" (K20) and later to the increase of "Profit" (K19). During the scenario evolution, raising profits will also reduce the "Number of trained staff" (K5) and increase "Labor productivity" (K7).

In the second scenario we start by decreasing the "Political stability" (K38). As an immediate effect, concepts "Customer income level" (K29), "Customer demand" (K28), "Stock prices" (K27), "Speed of adoption of new technology" (K3) will decrease. In the long term, the effect is most visible in another group of concepts: a decrease in "Labor productivity" (K7), "Employee loyalty" (K9), "Company reputation" (K33), and "Capital investments" (K46), as well as an increase in "Lost working time" (K6).

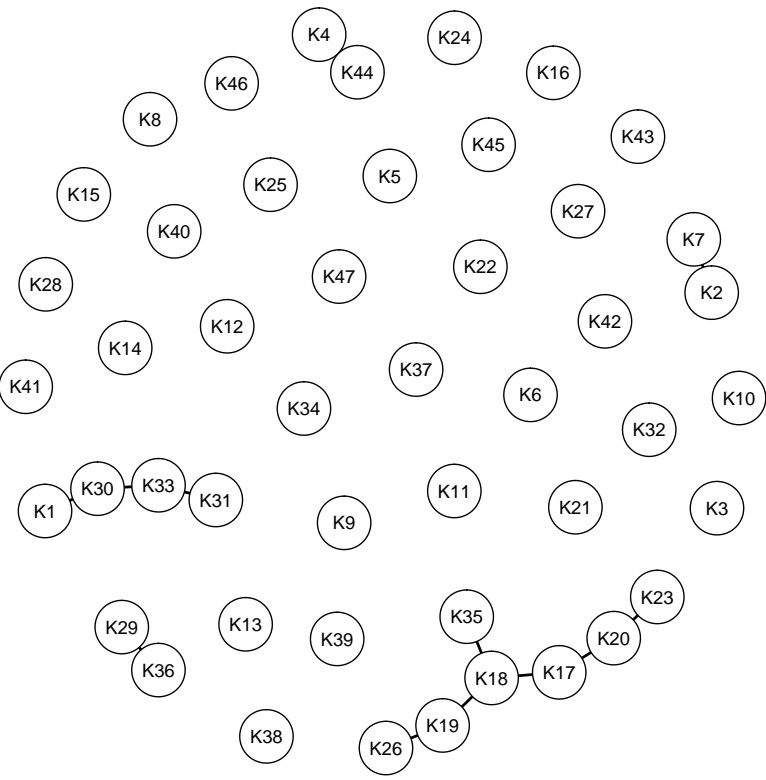

**Figure 5.** Mutual influence of system elements when using the composition *T*-norm *Algebraic product* and *S*-norm *Maximum*, and relevancy threshold of 0.7.

The third scenario, for which we present a detailed analysis, includes the effect of varying "Product quality" (K30) and "Employee loyalty" (K9) on "Customer loyalty level" (K32). We start by assessing the system's state change when only one of the controlling concepts—"Product quality"—is altered. As shown in Table 4, improving "Product quality" (K30) by itself does not have a significant impact on the "Customer loyalty level" (K32) until the fourth time step. This is possible due to an increase in the markup on goods and the potential rise in prices. If the states of both "Employee loyalty" (K9) and "Product quality" (K30) concepts were enhanced, the situation would change as shown in Table 5. This scenario has a considerably greater influence on the target concept and starts a loop of increased "Customer loyalty level" (K32), which will result in a boost to the company's overall state.

**Table 4.** Changes in the system concepts over five steps for the scenario of increasing "Product Quality" (K30).

| Concept | Step 0 | Step 1 | Step 2 | Step 3 | Step 4 | Step 5 |
|---|---|---|---|---|---|---|
| Employee loyalty | 0 | 0.04 | 0.07 | 0.12 | 0.22 | 0.38 |
| Staff turnover | 0 | 0 | −0.04 | −0.06 | −0.11 | −0.2 |
| Customer demand | 0 | 0.04 | 0.04 | 0.04 | 0.05 | 0.07 |
| Product quality | 0.1 | 0.1 | 0.12 | 0.14 | 0.18 | 0.25 |
| Customer service level | 0 | 0 | 0.01 | 0.02 | 0.02 | 0.06 |
| Customer loyalty level | 0 | 0.07 | 0.07 | 0.09 | 0.16 | 0.27 |
| Company reputation | 0 | 0.07 | 0.09 | 0.18 | 0.3 | 0.47 |
| Assortment of goods | 0 | 0 | 0 | 0.01 | 0.01 | 0.01 |
| Margin on goods | 0 | 0 | 0 | 0 | 0 | 0 |
| Profit | 0 | 0 | 0 | 0.15 | 0.08 | 0.16 |

**Table 5.** Changes in the system concepts over five steps for the scenario of increasing the concepts of "Product quality" (K30) and "Employee loyalty level" (K9).

| Concept | Step 0 | Step 1 | Step 2 | Step 3 | Step 4 | Step 5 |
|---|---|---|---|---|---|---|
| Employee loyalty | 0.1 | 0.14 | 0.24 | 0.36 | 0.59 | 0.94 |
| Staff turnover | 0 | −0.05 | −0.1 | −0.17 | −0.29 | −0.48 |
| Customer demand | 0 | 0.04 | 0 | 0.04 | 0.05 | 0.08 |
| Product quality | 0.1 | 0.1 | 0.14 | 0.2 | 0.28 | 0.43 |
| Customer service level | 0 | 0.03 | 0.05 | 0.1 | 0.18 | 0.29 |
| Customer loyalty level | 0 | 0.07 | 0.11 | 0.18 | 0.35 | 0.6 |
| Company reputation | 0 | 0.12 | 0.16 | 0.35 | 0.61 | 0.99 |
| Assortment of goods | 0 | 0 | 0 | 0.01 | 0.01 | 0.03 |
| Margin on goods | 0 | 0.05 | 0.07 | 0.09 | 0.14 | 0.2 |
| Profit | 0 | 0.01 | 0.02 | 0.21 | 0.18 | 0.35 |

The proposed retail system model allows the analysis of numberless managerial scenarios, as well estimation of the system's response to changing the activation value of one or more different concepts. Additionally, the probabilistic transitive closure matrix ($\mathbf{R}^*$) can be used to find the positive/negative pathways on a fuzzy graph, review required system adjustments, and identify concept co-dependencies.

## 5. Discussion

A retail business contains significant contradictions and implicit influences that obstruct target planning and assessing the impact of changes. One of the goals of this work was to offer a theoretical description of a retail system, modeling it as an FCM, and to provide a practical example of system analysis to the reader. Answering the first research question, in comparison to previous studies, the developed model includes a large number of relevant system concepts and links between them, including external economical and political factors. More specifically, views and perceptions of retail business stakeholders about the system's structure, connections, and main drivers were identified and incorporated into the model.

This study includes an extension to the fuzzy operator selection criteria described in reference [42]. This helped us to address the second research question, related to the choice of mathematical methods in FCM modeling. Fuzzy composition operator selection was shown to be an essential factor for improving the model. Therefore, we believe the proposed fuzzy operator selection rule can be applied in other FCM-related algorithms.

We provided a structural analysis of the model followed by three hypothetical examples of scenario modeling, expanding on previous research. For example, while previously developed retail business models [25] include core elements such as "Horizontal integration", "Vertical integration" and "Partners and networks", we extend this list with other influential concepts like "Customer service level", "Company reputation", "Product quality", and "Production standards". Further, in reference [34] the authors describe a potential scenario for a retail system, namely "unwillingness of managers to share information"; here, we provide additional scenarios such as "Increase of integration of systems with suppliers", "Decrease of political stability", and "Increase in product quality". Answering the third research question, scenario modeling allowed us to observe inverse or direct causal links, as well as degrees of change—using more concepts and distinct scenario setups than those presented in existing research.

The model can incorporate data sources such as financial reports (to track changes in the financial sub-sector), external economic indicators (to estimate inflation, exchange rates, and competition levels), and data from enterprise resource planning systems (to track suppliers and customers interactions). The capability of integrating data in this fashion

allows the model to consider quantifiable changes in real data, fostering the timeliness and quality of forecasts.

In contrast with several previous studies, we provide the model's open source code, as well as the data used to drive it. We hope this will facilitate future research on this topic and simplify the adaptation of the proposed methods for concrete use by retail companies. The software package allows addition or removal of concepts from the model, adjustment of expert estimations after additional interviews, or change of fuzzy operators in the modeling process. Therefore, adoption of this software in retail businesses can help companies reduce the time spent on scenario planning, and, in the long term, gain a competitive advantage. The developed software package can be used as a decision support system independent of the model area.

The proposed model is applicable to retail companies in developed markets, although limited to businesses operating in one country, due to the concepts such as "Political stability" (K38) and "Inflation expectations" (K39), which may be difficult to average for multinational companies. An additional constraint can be set on the size of the company; to avoid location bias, the modeled retail chain should contain at least 10 geographically dispersed shops without an e-commerce component. Another limitation of the model is the relatively small number of interviews. Additionally, no survey was conducted in order to analyze scenarios relevant to current strategies in operating companies due to the confidentiality of such information. Thus, the described scenarios were based on the author's best judgment.

## 6. Conclusions

In this paper, an FCM-based model of a retail system was proposed and a concrete case was presented using data gathered from expert domain knowledge and literature research. The fundamental concepts of retail systems were identified, and a fuzzy model was developed to aid in the business analysis. The impact of the driver concepts was illustrated and the worst and best case system development scenarios were compared. A software package with functions for fuzzy cognitive modeling, researching, and strategy monitoring in semi-structured systems was developed by the authors to produce the presented results, and made available as open source software.

The model can be used for the formulation and substantiation of management decisions in a retail organization, as well as to study, simulate, and test the impact of parameters on system behavior. The proposed model allows integration of the perspectives of various system stakeholders into the decision-making process. Experiments with a set of numerical examples revealed that the suggested strategic planning simulation mechanism can allow decision-makers to effectively develop robust strategic planning in time-variant competitive contexts. The model can be used by retail managers and policy makers to understand the impact of changes, assess their strategic importance, and identify interdependencies between different areas of the retail business.

Future development will likely focus on expanding the concepts of retail systems and integrating real-time measurable data into the model. Another area of focus might be the use of other learning algorithms to train the original FCM matrix. This may allow further calibration of the resulting model and identification of other hidden collateral connections, and has the potential to improve the system's explainability, as well as the results of scenario modeling.

**Author Contributions:** Conceptualization, A.P.; methodology, A.P.; software, A.P.; validation, A.P. and N.F.; investigation, A.P.; resources, N.F.; data curation, A.P. and N.F.; writing—original draft preparation, A.P. and N.F.; writing—review and editing, A.P. and N.F.; supervision, N.F. All authors have read and agreed to the published version of the manuscript.

**Funding:** This research was funded by the Fundação para a Ciência e a Tecnologia, Portugal under Grant No.: UIDB/04111/2020 (COPELABS).

**Institutional Review Board Statement:** Not applicable.

**Informed Consent Statement:** Not applicable.

**Data Availability Statement:** The data set used for this study is available at https://doi.org/10.528 1/zenodo.6046893, accessed on 16 February 2022.

**Conflicts of Interest:** The authors declare no conflict of interest. The funders had no role in the design of the study; in the collection, analyses, or interpretation of data; in the writing of the manuscript, or in the decision to publish the results.

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
