# Peer review of "Retail System Scenario Modeling Using Fuzzy Cognitive Maps"

_information, doi:10.3390/info13050251_

Round 1

Reviewer 1 Report

(1) Authors should clearly state in the manuscript what new knowledge this study intends to provide to the literature. Also, what is the difference between this study and related previous studies in the literature?

(2) The authors should explain the research question of this study in detail. Also, please develop a persuasive argument about the excellence of this study and the justification for publication.

(3) Systematic review and description of previous studies are very insufficient.

(4) Authors should clearly describe the research model and research methodology of this study. Also, please organize the arithmetic symbols used such as variables and parameters in a readable way. Authors also need to add detailed explanations about various formulas and mathematical expressions.

(5) In conclusion, the authors are requested to re-summarize the theoretical and practical contribution of this study. Also, please describe the implications and insights of this study. Also, please add the limitations of this study.

Reviewer 2 Report

This is an interesting paper that gives a detailed technical description of the fuzzy cognitive maps and the way they were adapted to the retail domain.

Some suggestions for the improvement of the paper are:

  • to develop more the arguments and the justifications for the FCM adaptation to the retail domain.
  • the discussion of the proposed model emphasize a lot on influencing factors, but discusses less the potential expected results and outcomes that want to be predicted by the use of the model.
  • what kind of managerial decision in the retail business can be substantiated by the proposed model? Better point out the envisaged (possible) outcomes of FCM scenario modelling. What is that you want to predict by the modelling? You pointed out strategy, but besides strategy (that is very generic), you need to include details on the final results that you refer to (to better illustrate the applicability of the model)
  • the paper offers a very good mathematical and model conception section, but it would be nice tot have more on the applicative part of the model in general terms.
  • for clarifications more details can be offered about the different scenarios used to test the proposed model and to conduct the structural analysis.
  • also it would be good to include a discussion on the type of real life data that could be used as input in the model in order to use it for managerial decision
  • at references the font is not the same and there are still inconsistencies in the style
  • more recent literature in the field (2019-2022) needs to be include in the paper. Most of it is much older literature. Classical is good, but new trends need to be included.

Round 2

Reviewer 1 Report

(1) Please rewrite the abstract and edit it into one paragraph. In abstract, I think it is desirable to write mainly about the results and meaning of the research rather than the background of the research.

(2) In the next round review process, please indicate the revised part in the updated manuscript. What the authors provide is a new manuscript with a  quite short points-by-points response. This alone makes it difficult to understand how the manuscript was effectively revised.

(3) In the introduction, authors should briefly describe the excellence and value of this paper and why it should be published. Also, please be more specific about FCM. In general, the explanation of research motivation and background in the introduction takes too much weight unnecessarily.

(4) In section 2, please describe systematically what the research on retail system was like in previous studies and what differences it is from this study. In addition, please describe systematically how the FCM-related research was and how it differs from this study on a topic other than the retail system.

(5) In section 3, it is difficult to understand why subsections 3.1 and 3.2 are necessary. FCM-related too basic information is introduced in detail and occupies a lot of pages. On the other hand, I don't think subsection 3.4 is worth breaking up into a separate subsection.

(6) Authors should describe in detail their research model and what they did in subsection 3.3. Also, the keywords presented in lines 338-354 do not seem to have much relation to the retail system. There is very little explanation as to why the authors chose this keyword and what it means. The authors reveal that they made a lot of modifications to the method in this round. However, the core contents of the study are not introduced very much, and non-core contents occupy too much weight.

(7) A simple introduction to the analysis results takes too much weight. On the other hand, interpretation and discussion of research results are very lacking, and should be presented along with a persuasive logical structure, intuitions and insights as an extension of previous research.

(8) Overall, it seems that the authors need to make a lot of effort to improve the quality of the writing in the manuscript.

Round 3

Reviewer 1 Report

I believe that the manuscript has been improved significantly. So, now the paper is very close to acceptance only except some minor issues.

(1) Authors are encouraged to create a new discussion section and add interpretation and discussion of the analysis results to this section. In addition, in the conclusion, the authors are requested to re-summarize the theoretical and practical contribution of this study and further describe the implications and insights of the study results.

(2) The font in Table 3 is small, making it difficult to read. Also, some pictures have a slightly lower resolution.
